# Characterization of Nano-SiO_2_/Zein Film Prepared Using Ultrasonic Treatment and the Ability of the Prepared Film to Resist Different Storage Environments

**DOI:** 10.3390/foods12163056

**Published:** 2023-08-15

**Authors:** Xiaofeng Ren, Junxia Wang, Arif Rashid, Ting Hou, Haile Ma, Qiufang Liang

**Affiliations:** 1School of Food and Biological Engineering, Jiangsu University, 301 Xuefu Road, Zhenjiang 212013, Chinawjxdyx0106@163.com (J.W.); arif_ktk007@hotmail.com (A.R.); houting19078607117@163.com (T.H.); mhl@ujs.edu.cn (H.M.); 2Jiangsu Provincial Key Laboratory for Physical Processing of Agricultural Products, Zhenjiang 212013, China; 3Institute of Food Physical Processing, Jiangsu University, 301 Xuefu Road, Zhenjiang 212013, China

**Keywords:** ultrasound, zein, SiO_2_, composite film, structure, mechanical properties

## Abstract

This study has developed, ultrasound-assisted, a novel food packaging film (U-zein/SiO_2_) for food packaging applications. Incorporating an optimal concentration of 18 mg/mL of nano-SiO_2_ and subjecting the film to 10 min of ultrasonic treatment resulted in a remarkable increase of 32.89% in elongation at break and 55.86% in tensile strength. In addition, the incorporation of nano-SiO_2_ effectively reduces the water content and solubility of the composite film, resulting in improved water/oxygen barrier properties. These physiochemical properties were further improved with the application of ultrasound. The analysis of attenuated total reflectance-Fourier transform infrared, X-ray diffraction, differential scanning calorimetry, and scanning electronic microscope demonstrated that the ultrasound treatment improved the hydrogen bonds, improved thermal stability, molecular arrangement, structure stability, and intermolecular compatibility of the composite film, resulting in enhanced physio-mechanical properties of the film. In addition, the ultrasound treatment led to a smoother film surface and reduced the pores on the film’s cross-section. Moreover, the U-zein/SiO_2_ film exhibited excellent mechanical and water/oxygen barrier properties in different storage environments over a period of 30 days. These results offer sound theoretical support for the practical application of the prepared preservative film.

## 1. Introduction

The preservation of food safety and quality, as well as the extension of shelf life, are critically dependent on the importance of appropriate packaging. Plastics are widely used in the food packaging industry due to their unique stability and low cost. However, plastics provide convenience to the industry, but also contribute significantly to environmental pollution. In addition, plastic packaging may release potentially harmful toxic substances to the human body [1]. Hence, the exploration of biodegradable, green packaging materials to replace traditional plastic packaging materials has emerged as a trend and focal point in the field of food packaging.

Zein is the main protein found in maize, accounting for about 45–50% of the total protein content. Zein, as a protein, possesses a considerable quantity of hydrophobic non-polar amino acids, resulting in a hydrophilic structure at both termini and a prominently hydrophobic structure at the lateral sides. Due to its unique structure, zein exhibits self-assembly characteristics, amphiphilicity, controllable particle size, and a wide range of isoelectric point [2]. Thus, zein can be effectively converted into different forms including films, fibers, microspheres, and nanoparticles. Moreover, it can interact with other natural polymer materials to form gels for the controlled delivery of bioactive substances [3]. Due to its film forming properties, zein has promising applications in the field of packaging as a structural material for preservative films. Studies have demonstrated that although zein-based film possess excellent water barrier properties, some inherent properties of films formed from single zein, such as poor processability, poor thermal stability, brittleness, and low elongation, limit their potential for use in flexible packaging [4]. Therefore, it is necessary to develop novel strategies for regulating the physicochemical properties of zein-based films.

In the past decade, there has been significant research and focus on the utilization of inorganic oxides, such as zinc oxide (ZnO), titanium oxide (TiO_2_), and silicon oxide (SiO_2_) particles, in combination with natural macromolecular preservative films, owing to the advancements in nanoscience and nanotechnology. SiO_2_ nanoparticles, commonly referred to as nano-SiO_2_, exhibit a three-dimensional structure and offer several advantageous properties, including cost-effectiveness, non-toxicity, and an amorphous powder state. Significantly, these nanoparticles exhibit notable characteristics, including a small particle size, a considerable specific surface area and energy, the presence of unsaturated chemical bonds, and the presence of hydroxyl groups on their surface [5]. The addition of nano-SiO_2_ has been reported to significantly improve the mechanical and thermal properties of alginate-based films [6]. Additionally, Luís et al. prepared sustainable alginate/collagen/SiO_2_ composite films and found that increasing the concentration of nano-SiO_2_ above 6% resulted in an approximately 8% reduction and 30% reduction in the water vapor permeability [7]. However, there are limited studies on the effect of nano-SiO_2_ on zein-based films. Therefore, a hypothesis arises proposing that the inclusion of nano-SiO_2_ possesses the capability to augment the physicochemical properties of zein-based films, thereby resulting in a comprehensive enhancement of their overall performance.

Indeed, the performance of zein-based films is influenced not only by the incorporation of nano-SiO_2_ but also by the specific preparation method employed for fabricating the films, making it a crucial factor to consider. Ultrasound, being a green non-thermal physical processing technology, has been widely used in various research and application fields of food, including extraction, enzymatic hydrolysis, emulsification, embedding, etc.; this is attributed to the three famous effects, the cavitation effect, mechanical effect, and chemical effect [8]. In recent years, ultrasound has been proven to be an effective tool in improving the performance of packaging films formed from natural macromolecular compounds. Eva et al., found that ultrasound treatment is an interesting strategy to improve the appearance of gluten film [9]. Some polypyrrole/SiO_2_ films elaborated under ultrasound irradiation were reported to exhibit the best protective performances [10]. Furthermore, ultrasound was introduced to assist in the preparation of nisin-loaded oat protein/pullulan films, and the results showed that ultrasound significantly increased the elongation at break, transparency, and surface morphology of the fabricated films [8]. Although there are few studies on the ultrasound-assisted preparation of zein-based film, our research group has carried out some research on other aspects of ultrasonic treatment of zein in previous years. For example, ultrasonic treatment of zein changed the particle size and uniformity of zein, increased the solubility and thermodynamic properties of the protein, thereby improving the enzymatic hydrolysis efficiency of zein [11,12]. Additionally, ultrasonic treatment of zein improved its ability to embed resveratrol. This phenomenon can be attributed to alterations in the secondary structure and spatial configuration of zein caused by the application of ultrasonic treatment. The treatment promotes the exposure of internal functional bonds such as hydrophobic interaction and disulfide bonds, thus strengthening the interaction between zein and resveratrol [2]. It was well known that during preparation of zein film using solvent casting method, the increase in solution polarity transforms the α-helix structure of zein into a β-sheet structure due to intermolecular and intramolecular interactions. Subsequently, the β-sheet structure is arranged in reverse with the head and tail connected to form a strip, which ultimately results in the formation of the film [3]. In general, the interaction between zein molecules, including hydrogen bonding, electrostatic interaction, hydrophobic interaction, van der Waals force, etc., directly affect the self-assembly behavior of zein, which is closely related to the structure and properties of the zein-based film. Considering the previous studies on ultrasound treatment of zein and the mechanism of zein film formation, we hypothesize that ultrasound could not only promote the uniform dispersion of nano-SiO_2_, but also improve the intermolecular interaction of zein, thereby affecting its self-assembly behavior and enhancing the performance of the final film.

The objectives of this study were to investigate the effects of ultrasonic treatment on the mechanical and physicochemical properties and structure of nano-SiO_2_/zein film. This research offers valuable insights into the fundamental mechanisms that underlie the preparation of nano-SiO_2_/zein films using ultrasonic assistance. It is worth mentioning that this study also specifically investigated the effects of different environmental conditions including relative humidity and temperature on the mechanical and physiochemical properties of the prepared nano-SiO_2_/zein film during storage. These investigations provide effective data and theoretical support for the practical application of the prepared preservative film.

## 2. Materials and Methods

### 2.1. Materials

Zein with a protein purity of 98% was obtained from Sigma-Aldrich (Steinheim, Germany). Nano-SiO_2_ with purity greater than 96% was purchased from Tianjin Longhua Chemical Co., Ltd. (Tianjin, China). All other reagents were of analytical grade. Deionized water was used in all the experiments.

### 2.2. Preparation of Zein/SiO_2_ Film

The zein/SiO_2_ films were prepared using a solution casting method. Briefly, zein was fully dissolved in 80% (*v*/*v*) ethanol aqueous solution to prepare 0.14 g/mL zein solution. Then, a complex of glycerol, polyethylene glycol 400, and oleic acid with a mass ratio of 1:1:2 was added as a plasticizer, and its final concentration in the mixed solution was 56 mg/mL. After 30 min of magnetic stirring, nano-SiO_2_ powder was slowly added at a concentration of 9–45 mg/mL. The zein/SiO_2_ film-forming solution was prepared through magnetic stirring for a duration of 15 min. Subsequently, a volume of 10 mL of the film-forming solution was applied onto a disposable plastic plate for a period of 10 min, followed by drying at 60 °C for 2 h. For subsequent experiments, the prepared films were placed inside a desiccator, which was filled with saturated K_2_CO_3_ solution and maintained at an ambient temperature of 25 °C and a relative humidity of 43%. It is worth mentioning that the preparation method for the zein-loaded film was identical to the aforementioned process, with the only difference being the addition of nano-SiO_2_.

### 2.3. Ultrasound-Assisted Preparation of Zein/SiO_2_ Film

The freshly prepared zein/SiO_2_ film-forming solution was securely sealed within high-pressure resistant plastic bags to undergo multi-frequency power ultrasound treatment. The ultrasonic equipment was developed by our group and manufactured by Meibo Biotechnology Co., Ltd. (Zhenjiang, China). The ultrasonic conditions used were as follows: ultrasound power density, 150 W/L; time, 0–25 min; pulsed on-time/off-time, 5 s/2 s; frequency, 60 kHz; and ultrasound temperature, 25 °C. The film-forming solution, following ultrasonic treatment, was utilized to create a film using the same methodology as described in Section 2.2.

### 2.4. Determination of Film Properties

#### 2.4.1. Mechanical Properties

The tensile strength (TS) and elongation at break (EBA) were measured by a texture analyzer (TA-XT2i, Stable Micro Systems, Godalming, UK). The operating parameters of the device were set as follows: the speed before the test, 1.00 mm/s; the test speed, 2.00 mm/s, the post-test speeds, 10 mm/s, the trigger force, 5 g, and the initial clamping distance, 30 mm.

#### 2.4.2. Moisture Content, Swelling Degree, and Solubility

The moisture content of the films sample was determined using the method of Kang et al. [8] and was reported as the percentage of water loss in the total mass of the film after drying to constant weight.

The films sample was cut into small pieces of 3 × 2 cm, dried at 50 °C to constant weight (*m*_1_) and immersed into 50 mL distilled water. The water was replaced every 24 h for 5 days, and the film was weighed and named *m*_2_. Then, it was placed in a 50 °C oven until it had a constant weight (*m*_3_). The calculation formulas of the swelling degree and solubility were as follows:(1)$$\mathrm{Swelling}\;\mathrm{degree}\left(\%\right)=\frac{\left(m_{2}-m_{3}\right)}{m_{3}}\times 100$$
(2)$$\mathrm{Water}\;\mathrm{solubility}\left(\%\right)=\frac{\left(m_{1}-m_{3}\right)}{m_{1}}\times 100$$

#### 2.4.3. Water Vapor Permeability (WVP) and Oxygen Permeability (OP)

Standard test method ASTM E96 was used for the measurement of WVP of the film samples with some modifications [13]. After cutting, a film sample with an area of 12.56 × 10^−4^ m^2^ was obtained, and its thickness was measured. The film was sealed on the beaker opening with anhydrous calcium chloride by Vaseline. The sealed beakers were then transferred to a desiccator containing saturated potassium nitrate solution (relative humidity 90%). The above beaker was weighed every 2 h at room temperature for 12 h. The WVP was obtained by linear regression of mass increment and time, and its calculation formula was as follows:(3)$$\mathrm{WVP}=\frac{\Delta m\cdot d}{A\cdot \Delta P\cdot T}$$
where *d* is the thickness of composite films; Δ*m* is the water increment transferred through the film surface in *T* time, g; *A* is the exposed composite film area; Δ*P* is the water vapor pressure difference between the two sides of composite films; and *T* is the interval time of weighing, s.

A 250 mL conical flask was filled with 25 mL of soybean oil before the film sample was sealed at the mouth of the bottle and a thin layer of white vaseline was equally spread around the bottle. The peroxide value (POV) of soybean oil was determined with the Na_2_S_2_O_3_ titration method after being placed at 60 °C for 7 days to calculate the OP of the film [14]. The following formula was employed to calculate the POV of the film:(4)$$\mathrm{POV}\;(\mathrm{mg}/\mathrm{kg})=\frac{C\times \left(V-V_{0}\right)}{m}\times 1000\;$$
where C is the normality of Na_2_S_2_O_3_, mol/L; V and V_0_ were the volume of sodium thiosulfate consumed in the sample and blank test, mL, respectively; and m is the sample mass, g.

### 2.5. Structural Characterization of Films

#### 2.5.1. Attenuated Total Reflectance-Fourier Transform Infrared (ATR-FTIR) Spectroscopy

An ATR-FTIR spectrometer (Q2000-DSC, TA Instruments, New Castle, DE, USA) was used to record the FTIR spectra of the film sample, with a resolution of 4 cm^−1^ and a wavenumber range of 650–4000 cm^−1^. The spectra of the samples were also processed and the secondary structure of the protein in the film was quantified.

#### 2.5.2. X-ray Diffraction (XRD)

X-ray diffractometer (Shimadzu 6100, Kyoto, Japan) was used for the determination of the film patterns at a scanning speed of 5°/min in the range of 2θ from 5° to 80°.

#### 2.5.3. Differential Scanning Calorimetry (DSC)

The dried sample was sealed in an aluminum plate and weighed. The sealed samples were then loaded into a differential scanning calorimeter (Q2000-DSC, TA Instruments, New Castle, DE, USA). In an inert nitrogen atmosphere, the sample was scanned during heating from 25 °C to 300 °C at a rate of 10 °C min^−1^.

#### 2.5.4. Scanning Electron Microscopy (SEM)

The cross-section samples were prepared by freezing the dried composite film with liquid nitrogen, and then the samples were placed on an electron microscope for vacuum gold plating. The film samples surface and cross-section morphology were determined with scanning electron microscope (FEI NovaNano450, Hillsboro, OR, USA) at 10,000× and 30,000× magnification.

### 2.6. Statistical Analysis

Each experiment was repeated a minimum of three times to ensure reliability, and the results were presented as mean values accompanied by their respective standard deviations. Statistical analysis of the data was performed using SPSS 26.0 software, with significant differences among the means determined through Duncan’s test and independent two-sample *t*-test, employing a significance level (*p* < 0.05). (SPSS Inc., Chicago, IL, USA). Graphs and visual representations were generated using OriginPro 2023 software (OriginLab Co., Ltd., Northampton, MA, USA).

## 3. Results and Discussion

### 3.1. Influence of Ultrasonic Treatment on Physico-Mechanical Properties of the Films

#### 3.1.1. Single-Factor Optimization Focusing on the Mechanical Properties

Mechanical properties such as TS and EBA play a significant role in the quality determination of the biodegradable. The TS and EBA of ultrasound-assisted nano-SiO_2_/zein film were determined. Effects of nano-SiO_2_ addition and ultrasound treatment on TS and EAB of films were investigated (Figure 1). The addition of 18 mg/mL of SiO_2_, and other doses of SiO_2_ significantly reduced the TS of the fabricated film, while with the increase in SiO_2_ addition, the EBA of the film initially increased and then decreased. At a SiO_2_ concentration of 18 mg/mL, the film exhibited the highest EBA compared to the control group, while the TS did not show significant changes. The results showed the percentage of SiO_2_ added significantly affected the mechanical properties of the composite film. Our findings are consistent with the outcomes reported by Luís et al., who developed a film based on nano-SiO_2_ and sodium alginate [6]. The optimal concentration of SiO_2_ can effectively adsorb and form bonds with zein, ultimately filling the gaps present in the three-dimensional network structure of zein, thus enhancing the mechanical properties of the film. However, an excessive amount of SiO_2_ proved challenging to uniformly distribute within the film, leading to particle agglomeration that could potentially destabilize the composite film’s structure and hinder its capacity to withstand external stress, ultimately resulting in a significant decline in its mechanical properties. The impact of ultrasonic treatment duration on the mechanical properties of the composite film aligns with the observations made for varying nano-SiO_2_ concentrations. It is noteworthy that a 5-min duration of ultrasonic treatment considerably decreased the TS and EBA parameters of the composite film, indicating that short-term ultrasonic treatment can potentially disturb the original arrangement of nano-SiO_2_/zein within the structure of the composite film. Upon applying 10 min of ultrasound treatment, the composite film exhibited a TS of 10.87 MPa and EBA of 87.07%, as the ultrasound waves caused the nano-SiO_2_/zein components to form a more stable structure, resulting in an improvement in mechanical properties. In comparison to the control group, the composite film exhibited a significant (*p* < 0.05) increase in both TS and EBA, with values rising by 32.89% and 55.86%, respectively. This may be because relatively long-term ultrasound destroyed the structure of zein and exposed more interaction sites, which increase intermolecular interactions such as the H-bond interaction between zein and nano-SiO_2_. In addition, micro jets, the shear forces and shock waves induced by ultrasound, will reduce the SiO_2_ molecular weight of and enhance its uniformity in the film-forming solution, thereby improving the mechanical properties of the composite film [10]. However, with an ultrasound time greater than 15–25 min, the trend of TS and EBA of the composite film was reversed. The results showed that the structural and intermolecular changes of zein caused by excessive ultrasonic treatment were not suitable for improving the mechanical properties of nano-SiO_2_/zein films. In conclusion, the results indicated that different durations of ultrasound treatment result in distinct changes in TS and EBA of the composite film. These changes may arise from the dynamic changes in protein structure and intermolecular forces caused by ultrasound treatment [15].

#### 3.1.2. Moisture Content, Swelling Degree and Solubility

The evaluation of the sensitivity of preservative film to water includes the determination of water content, swelling degree, and solubility, which seriously affects the biodegradability of the film and the quality of packaged food [16]. As shown in Table 1, the moisture content of zein and zein/SiO_2_ film were 17.49% and 16.42%, respectively. The findings revealed that the incorporation of SiO_2_ led to a decrease in the moisture content of the zein film. Hou et al., obtained a similar result, demonstrating that the inclusion of nano-SiO_2_ decreased the moisture content of agar/sodium alginate film; the reduction was attributed to the interaction between nano-SiO_2_ and polysaccharides, which included hydrogen bonding, thereby limiting the contact between water molecules and the hydrophilic groups within the nanocomposite films [5]. Ultrasonic treatment resulted in a further reduction in the moisture content of Zein/SiO_2_ film to 15.65%. This treatment is believed to have changed the structure of zein, facilitated the dispersion of nano-SiO_2_ particles in the film-forming solution, and increased the network crosslinking of the film-forming solution, leading to the decrease in water content. It was observed that the swelling degree and solubility of the zein film were 97.67% and 17.49%, respectively, while for the zein/SiO_2_ film, these values were 112.93% and 24.4%, respectively. The obtained result clearly indicated that the incorporation of nano-SiO_2_ into the film substantially enhanced the swelling degree and solubility of the composite films. The swelling degree is closely related to the inner micro-structures of the films, and the solubility is used to evaluate the escape ability of the material from the composite films. Based on the findings, it can be inferred that the introduction of nano-SiO_2_ altered the internal structure of the composite film, leading to an enhanced dissolution ability of the membrane matrix. Ultrasonic treatment increased the swelling degree of the composite film to 120.65%, while the solubility decreased to 22.62%. The ultrasound treatment may have increased the cavity volume inside the composite film molecules, resulting in a solid molecular structure that provided the composite film with increased water holding capacity. Furthermore, the reduction in the solubility of the film, which could be attributed to the decrease in the water solubility of the film matrix, emphasizes an improvement in the integrity of the prepared film when exposed to water, thereby enhancing its stability and durability. The alteration in zein structure, attributed to the ultrasonic treatment, resulted in the exposure of additional hydrophobic groups on the film’s surface. This phenomenon could account for the observed changes in the film’s properties [12].

#### 3.1.3. WVP and OP

The process of water vapor and oxygen penetrating the composite films can be divided into three stages: adsorption, diffusion, and decomposition. The high WVP of the film will cause a large loss of water in the packaged food; the high OP will accelerate the oxidation of food and shorten the shelf life of food. Therefore, to fully comprehend the properties of the films, it is pivotal to conduct a thorough analysis of their WVP and OP. According to the data presented in Table 1, the zein-loaded film exhibited a WVP of 13.07 × 10^−11^ g/m·s·Pa and a POV of 144.97 mmoL/kg. The addition of nano-SiO_2_ significantly (*p* < 0.05) decreased the WVP and POV to 7.21 × 10^−11^ g/m·s·Pa and 99.44 mmoL/kg. Similar to Hou et al.’s findings, the incorporation of 2.5 wt% nano-SiO_2_ in agar/sodium alginate films led to notable enhancement in the water vapor barrier properties of the films, as indicated by a substantial reduction in WVP [5]. The reason for this might be attributed to the strong Interaction between the membrane matrix that facilitated the formation of a denser structure. Furthermore, the high aspect ratio of SiO_2_ nanoparticles, which were dispersed throughout the membrane matrix, act as impermeable barriers. Water molecules attempting to penetrate the film must negotiate a tortuous path around these nanoparticles [6]. 

Consequently, the path that water molecules take through the film was prolonged, ultimately resulting in reduced water vapor transmittance. Moreover, after ultrasound treatment, the WVP value of the composite film was significantly (*p* < 0.05) reduced to 6.59 × 10^−11^ g/m·s·Pa. On one hand, ultrasonic treatment is believed to break apart partially aggregated nano-SiO_2_, effectively dispersing it evenly throughout the membrane matrix, thus maximizing its water-blocking effect. On the other hand, ultrasound may increase the non-covalent interaction and crosslinking degree between the membrane matrix by changing the molecular structure of zein. This structural change results in a more compact membrane matrix that impedes the permeation and evaporation of water vapor, effectively enhancing the water vapor barrier properties of the material [17,18]. Notably, the POV of Zein, Zein/SiO_2_ and U-Zein/SiO_2_ films were144.97, 99.44 and 87.64 mmoL/kg, respectively. The effects of nano-SiO_2_ and ultrasound on the OP and WVP of the film were similar. In conclusion, the above results indicated that the addition of nano-SiO_2_ and ultrasonic treatment significantly reduced the OP and WVP of the composite film. It could also be further speculated that the U-Zein/SiO_2_ composite film prepared could effectively delay the metabolism of aerobic microorganisms, inhibit food oxidation and rancidity, and solve the quality loss of fresh fruits due to water loss to a certain extent.

### 3.2. The Alterations in the Interactions and Structure of Films Caused by Exposure to Ultrasound

#### 3.2.1. ATR-FTIR Spectra

ATR-FTIR has proven to be an effective method of examining the interaction of natural macromolecules. The FTIR spectra of the composite films are shown in Figure 2. The distinctive spectral peaks corresponding to the presence of zein within the film were identified at specific wavenumbers: 1541 cm^−1^ (indicative of amide II, N-H stretching vibration), 1654 cm^−1^ (characteristic of amide I, C=O stretching vibration), 2965 cm^−1^ (representing C–H stretching vibration, including -CH2 and -CH3 bending vibrations), and 3302 cm^−1^ (associated with O–H and N–H stretching vibrations) [19,20]. Although the FTIR spectra of zein/SiO_2_ and U-zein/SiO_2_ films are similar to that of zein-loaded film, a closer look still revealed some differences; a new absorption peak appeared at 1100 cm^−1^, which is characteristic peaks of nano-SiO_2_, representing asymmetric stretching vibration of Si-O-Si [5]. Furthermore, the addition of nano-SiO_2_ made the peaks representing the C-H, O-H, and N-H stretching vibration of the composite films blue-shifted to 2931 cm^−1^ and 3298 cm^−1^, respectively, indicating the existence of hydrogen bonds between SiO_2_ and zein. Likewise, the characteristic peak of amide I in zein/SiO_2_ film appeared at 1649 cm^−1^ instead of 1654 cm^−1^, suggesting that the intervention of SiO_2_ changed the secondary structure of zein to some extent. Compared with zein/SiO_2_ film, some characteristic peaks of FTIR spectra of U-zein/SiO_2_ films were shifted, and the intensity of some characteristic peaks was enhanced. Following ultrasonic treatment, the peak representing amide II in the composite film shifted from 1541 cm^−1^ to 1539 cm^−1^. These findings suggest that ultrasonic treatment led to further alterations in the secondary structure of zein within the zein/SiO_2_ film. Jin et al. used dual-frequency power ultrasound to treat corn gluten meal and found that ultrasound changed the secondary structure of zein; ultrasound reduced the content of α-helix and β-turn in zein by 3.3% and 3.8%, respectively, while β-folding increased by 8.2% and random coil increased by 4.2% [21]. The application of ultrasound treatment to zein induces structural modifications, which consequently result in alterations in the exposed interaction bonds of zein. These changes have the potential to enhance the intramolecular interactions within zein and promote interaction forces between zein and SiO_2_, including the formation of hydrogen bonds. This speculation is supported by the shifts in the C-H, O-H, and N-H stretching vibration peaks of U-zein/SiO_2_ films to 2928 cm^−1^ and 3303 cm^−1^, respectively. The aforementioned findings provide additional insight into the underlying factors contributing to the notable enhancements in mechanical properties and the film’s capability to effectively hinder the permeation of water vapor and oxygen subsequent to ultrasonic treatment.

#### 3.2.2. XRD

Figure 3 illustrates the XRD spectra of the zein, zein/SiO_2_ and U-zein/SiO_2_ films. The XRD pattern of the zein film shows two broad diffraction peaks at 2θ = 12.24° and 2θ = 19.57°, respectively, indicating that the film has an amorphous structure. The zein/SiO_2_ and U-zein/SiO_2_ films exhibited similar XRD pattern compared to zein film. As pure nano-SiO_2_ was reported to be an amorphous material with just a broad diffraction peak at 2θ ≈ 23° [6]. It could be speculated that ano-SiO_2_ was successfully encapsulated by zein into the interior of the network structure during the casting process, resulting in its own characteristic peaks that cannot be detected. It was also found that the addition of SiO_2_ made the characteristic peaks of zein move to 13.34° and 19.03°; the intensity of the peak at 13.34° was significantly enhanced; and the crystallinity of the film was correspondingly significantly increased from 39.09% to 52.72%. The results show that the addition of SiO_2_ induced the formation of a more ordered structure of the membrane matrix, and the molecular interaction was enhanced. Following the application of sonication, the diffraction peaks experienced an additional shift towards 13.14° and 19.24°, accompanied by a remarkable reduction in both the width and intensity of these peaks. Consequently, the crystallinity of the U-zein/SiO_2_ film decreased significantly to a value of 43.51%. Kang et al., also found that ultrasonic treatment could significantly reduce the crystallinity of Pullulan/Oat protein/Nisin composite film; the prepared composite film exhibits more amorphous features [8]. It is a widely acknowledged fact that the propagation of ultrasound in a medium leads to the generation of numerous cavitation bubbles. When the negative pressure within these bubbles surpasses the surface tension of the medium, the bubbles expand and eventually collapse, resulting in the generation of powerful shear forces and turbulence. This collapse process is accompanied by a significant rise in local temperature (~5000 K) and pressure (~1000 bar) [22]. The highly intense environment generated by ultrasound, as described earlier, has the potential to disrupt the preexisting ordered structure of the film-forming matrix. This disruption, in turn, facilitates the re-exposure of internal interaction bonds within the matrix, thereby creating favorable conditions for the aggregation and rearrangement of macromolecular substances [23]. Consequently, ultrasonic treatment has the potential to induce the formation of a zein/SiO_2_ film with an entirely distinct structure, diverging from its original composition.

#### 3.2.3. Thermal Stability

The DSC and thermogravimetric analysis (TGA) curves obtained from the film samples provide valuable data to determine key thermodynamic parameters, namely T_0_ (onset temperature), Tp (peak temperature), and Tc (end temperature). These parameters offer insights into the thermal behavior and stability of the film samples. Typically, the weight loss observed in the film can be categorized into three distinct stages based on the thermodynamic parameters. In the first stage, occurring from ambient temperature to T_0_, the film undergoes the removal of surface-bound water and small molecules like glycerol. Subsequently, during the temperature range between T_0_ and Tp, the decomposition of zein’s side chains takes place. Lastly, from Tp to Tc, the main chain of zein within the film matrix undergoes complete dissociation [24]. As shown in Table 2, the ranges T_0_, Tp of zein and zein/SiO_2_ film were 79.57 °C, 204.90 °C and 82.36 °C, 208.37 °C, respectively; compared with zein film, the range T_0_, Tp of zein/SiO_2_ film increased significantly. The findings of the study showcased that the inclusion of SiO_2_ in the film exhibited a notable retardation in film dehydration and enhanced the resistance to disruption of the spatial network structure and intermolecular non-covalent bonds within the film. As a result, the thermal stability of the film was significantly improved [25]. As compared to zein film, the significant increased Δ*H* of zein/SiO_2_ film also confirmed the above speculation. Ultrasound treatment further increased the Tp of zein/SiO_2_ film to 219.59 °C, indicating that ultrasound helped to form a denser network structure and enhanced intermolecular interactions between the film substrates, especially the hydrogen bonds among polymers, glycerol, and water as described by ATR-FTIR (Figure 2). While SiO_2_ and ultrasound did not change the T_0_, Tc, and ΔH of the film. The reason may be that neither SiO_2_ nor ultrasound had some effect on chemical structure of the film-forming matrix and the covalent bonds within the molecules. Theoretically, the thermal effect (~5000 K) induced by ultrasound and the generation of local free radicals have the potential to disrupt the chemical structure of natural macromolecules. However, it is important to note that the exceedingly high temperatures resulting from bubble breakup are confined to the collapse of bubble nuclei and limited regions near the surface of the bubble [26]. The circulating water used in this study could effectively control the ultrasonic treatment temperature within a certain range.

#### 3.2.4. SEM

The surface and cross-section morphologies of the nanocomposite films were examined using a scanning electron microscope (with magnifications of 10,000× and 30,000×). Experimental verification confirmed that the incorporation of SiO_2_ and the application of ultrasonic treatment led to distinct observable variations in both the surface and cross-section of the sample (Figure 4). The zein-loaded film showed a relatively flat, uniform surface; it was attributed to the ordered self-assembly of zein molecules. Comparatively, the cross section of zein-loaded film exhibited some round larger pores, which could contribute to the penetration of gases including water vapor and oxygen. In contrast, the surface of the zein/SiO_2_ film exhibited the presence of small particles and, in some cases, large aggregates. Under a magnification of 30,000×, the surface of the zein/SiO_2_ film displayed a markedly rough and uneven texture. The results showed that although SiO_2_ could improve the mechanical properties of the film to a certain extent, the addition of SiO_2_ gave the film a poor morphology [5]. From another point of view, it could also be inferred from the apparent morphology of the film that there was not a simple physical mixing between SiO_2_ and zein, and there was some intermolecular interaction between them, which has been proved by the results of ATR-FTIR and XRD. Furthermore, in comparison to the zein-loaded film, the cross-section analysis of the zein/SiO_2_ film revealed a decrease in the number of voids, accompanied by the presence of fillers within these voids. This observation provides an explanation for the lower water vapor permeability (WVP) and oxygen permeability (OP) exhibited by the zein/SiO_2_ film when compared to the zein-loaded film. At the same time, the ultrasound treatment reduced the surface roughness and discontinuity of the zein/SiO_2_ film and further reduced the size and number of voids on the cross section. The results demonstrate that ultrasonic treatment improves the compatibility between nano-SiO_2_ and zein. The prepared U-zein/SiO_2_ film exhibits a smooth, continuous, and compact surface. Similarly, Cheng et al., reported that high-intensity ultrasound could reduce cracks and protein aggregates on the surface of pea protein films. As a result, the transparency and tensile strength of the film improved significantly, and its water content and water vapor permeability decreased [27]. The cavitation effect induced by ultrasound not only enhanced the dispersion of nano-SiO_2_ but also induced certain alterations in the macromolecular structure of zein. These changes were conducive to exposing hydrophobic residues and non-covalent active sites, thereby facilitating improved interactions between macromolecules. Moreover, the cavitation effect led to a reduction in crystallite size and friction coefficient during the film-forming process [8]. As a result, it could be speculated that ultrasonic treatment could improve the mechanical and physiochemical properties of the film by affecting the microstructure of the film.

### 3.3. The Ability of U-Zein/SiO_2_ Film to Resist Different Storage Environments

As a food packaging material, biological matrix film must typically withstand a range of environmental conditions, including storage time, ambient temperature, and humidity. Therefore, it is necessary to evaluate the impact of various environmental pressures on the mechanical and physicochemical properties that closely relate to the preservation effect of composite films.

#### 3.3.1. Ambient Humidity

The effects of different humidity on EBA, TS, WVP, and OP of the film during 30 days of storage were investigated. As shown in Figure 5A,B, with the increase of storage time, the WVP and OP of the film showed a rapid increase (0–6 d), and then remained basically unchanged (7–30 d). The observed phenomenon can be attributed to the gradual accumulation of water on the film’s surface during the initial 6 d of storage, resulting in the absorption of water and subsequent swelling of the film’s surface. This led to a loosening of the internal structure of the film, which in turn facilitated the diffusion of water vapor and oxygen, resulting in decreased water and oxygen barrier properties of the composite film. Lower ambient humidity levels were associated with lower WVP and OP of the film, indicating improved water and oxygen resistance. Especially for the film in the environment with a RH 75%, in order to balance with the ambient humidity, the water absorption rate of the film might be the fastest and the water absorbed was the most; as a result, the WVP and OP values reached the highest compared to other humidity films. After 30 d of storage, there was no significant difference in WVP and OP values between RH 43% films and RH 54% films. Therefore, from the perspective of WVP and OP, the optimal RH for storage is 43%. After 30 days of storage at 43% ambient humidity, the WVP and POV of the film increased from 6.59 × 10^−11^ g/m·s·Pa, 87.64 mmol/kg to 11.22 × 10^−11^ g/m·s·Pa, 135.28 mmol/kg, respectively.

As shown in Figure 5C, the EBA of the 54%- and 75%-humidity films demonstrated an increasing trend during the first 6 d of storage, followed by a subsequent decrease during days 7–30. Conversely, the EBA of the 43%-humidity film displayed the opposite trend during the first six days of storage. The film used in this study was initially stored in a 43%-humidity dryer, meaning the film would not absorb water in an environment of 43% humidity. However, in environments with 54% and 75% humidity, the film would rapidly absorb water. The increase in appropriate water content is beneficial for improving EBA. Additionally, the reason for the decrease in EBA of the film caused by the storage time of more than 6 days was due to the negative effect of excessive moisture absorption on EBA of the film. In addition, the ambient light and oxygen may also change the structure of the film and affect EBA. Compared with EBA, the TS of the film showed the opposite trend; as the storage time increased, the TS of the film in all humidity environments gradually increased (Figure 5D). Thus, selecting an appropriate ambient humidity level is necessary to improve the mechanical properties of the film during storage. Based on the main consideration of EBA, the optimal ambient relative humidity was found to be 54% throughout the storage process. After 30 days of being stored at an ambient humidity of 54%, the EBA of the film decreased from 108.35% to 88.37%, while its TS increased from 5.53 MPa to 10.87 MPa.

#### 3.3.2. Ambient Temperature

Freezing temperature (−20 °C), cold storage temperature (4 °C), and room temperature (25 °C) are crucial temperatures for food storage. Therefore, it is essential to study the EBA, TS, WVP, and OP of the film at these temperatures. As depicted in Figure 6A,B, the WVP and OP of the film increased with storage time, and this increase was more pronounced at 4 °C > −20 °C > 25 °C. The film stored at 25 °C had the least impact on its WVP and OP. On the other hand, 4 °C and −20 °C ambient temperatures decreased the barrier ability of the film towards water vapor and oxygen to varying degrees. This phenomenon could be attributed to the impact of low temperature on the spatial molecular structure of zein, causing a reduction in intermolecular interaction forces within the film, potentially affecting its physiochemical properties. Moreover, when the temperature of the film dropped to −20 °C, the liquid water in the film transformed into solid water, which could have an impact on the distribution and morphology of water in the film matrix, leading to a reduction in gas channels inside the material. As a result, the film stored at −20 °C had better gas-blocking ability than the material stored at 4 °C. Considering the WVP and OP, the U-zein/SiO_2_ film was recommended for use at 25 °C, followed by −20 °C. After 30 d of storage at 25 °C, the WVP and OP of the film increased from 6.59 × 10^−11^ g/m·s·Pa and 87.63 mmol/kg to 11.35 × 10^−11^ g/m·s·Pa and 138.19 mmol/kg, respectively.

The effect of storage temperature on the mechanical properties of the films was exhibited in Figure 6C,D. With the extension of storage time, the EBA of the composite films showed a decreasing trend, which might be related to the migration of water during storage. Compared with other temperatures, the EBA of the film stored at −20 °C was the lowest. The reason might be that the ambient temperature of −20 °C changed the ordered structure of zein, and the interaction force between zein and SiO_2_, such as hydrogen bond, also decreased to varying degrees. Unlike EBA, with the increase in storage time, especially in 7–30 days, the TS of the film increased gradually at all different temperature environments. During the 18 d of storage, the TS of the film was the highest at −20 °C compared to other temperatures. The TS of the films stored at −20 °C and 4 °C was significantly higher than that at 25 °C during 18–24 d. After 30 d of storage, there was no significant difference in TS between the films at the three test temperatures. Based on the analysis of EBA and TS of the film, it is recommended to store the prepared film at 4 °C when the storage time exceeds 24 days. After being stored at 4 °C for 30 days, the EBA of the film decreased from 108.35% to 60.41%, while its TS increased from 5.53 MPa to 12.68 MPa.

## 4. Conclusions

This study focuses on the development of a novel active degradable packaging film made of U-zein/SiO_2_. The study investigates the influence of ultrasonic duration and nano-SiO_2_ concentration on the mechanical properties of the composite film. The optimal ultrasonic time and nano-SiO_2_ concentration were found to be 10 min and 18 mg/mL, respectively. The study’s findings indicated that incorporating nano-SiO_2_ significantly reduced the water content, solubility, WVP, and OP of the composite film, while enhancing its swelling degree. Furthermore, the mechanical and physicochemical properties of the composite films were improved with the application of ultrasound intervention. FTIR analysis revealed that ultrasound treatment further enhanced the formation of hydrogen bonding between nano-SiO_2_ particles and the matrix molecules. XRD analysis indicated that the application of ultrasonic treatment resulted in a decrease in the crystallinity of the composite film from 52.72% to 43.51%. Furthermore, the U-zein/SiO_2_ film exhibited a greater degree of amorphous characteristics. Furthermore, nano-SiO_2_ and ultrasound treatment could both improve the thermal stability of the composite film. SEM showed that ultrasound promoted nano-SiO_2_ dispersing homogeneously in the films and reduced the size and number of pores on the cross section of the film, and the prepared film showed a relatively smooth, continuous, and compact surface. In addition, the effect of different relative humidity (43% RH, 54% RH and 75% RH) and ambient temperature (−20 °C, 4 °C and 25 °C) on the mechanical properties and water/oxygen barrier ability of U-zein/SiO_2_ film during 30 days of storage was further investigated. The study’s results demonstrate that ultrasound is an eco-friendly and simple processing technology that can enhance the mechanical and physiochemical properties of the macromolecular biodegradable film. Nevertheless, further investigation is necessary to explore the actual preservation impact of the U-zein/SiO_2_ film on food.

## Figures and Tables

**Figure 1 foods-12-03056-f001:**
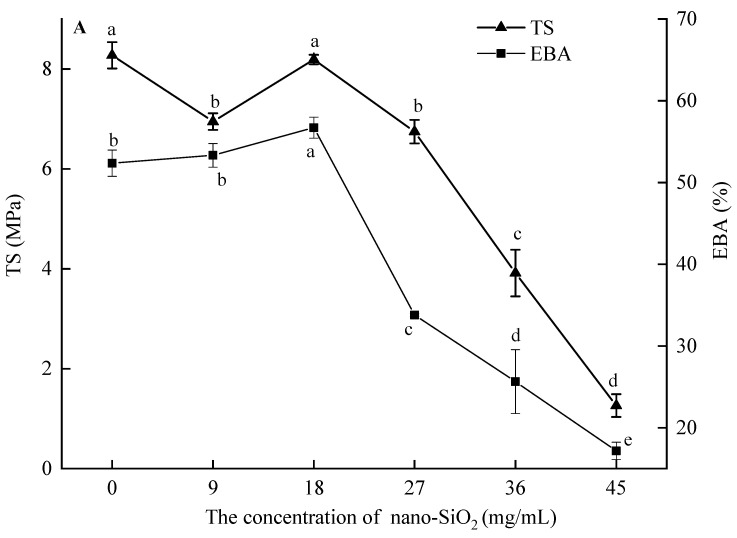
The effects of nano-SiO_2_ concentration (**A**) on the elongation at break (EBA) and tensile strength (TS) of the composite film without ultrasonic treatment; the effect of ultrasonic treatment time (**B**) on EBA and TS of composite films. Different lowercase letters in the same index indicate significant differences between groups (*p* < 0.05).

**Figure 2 foods-12-03056-f002:**
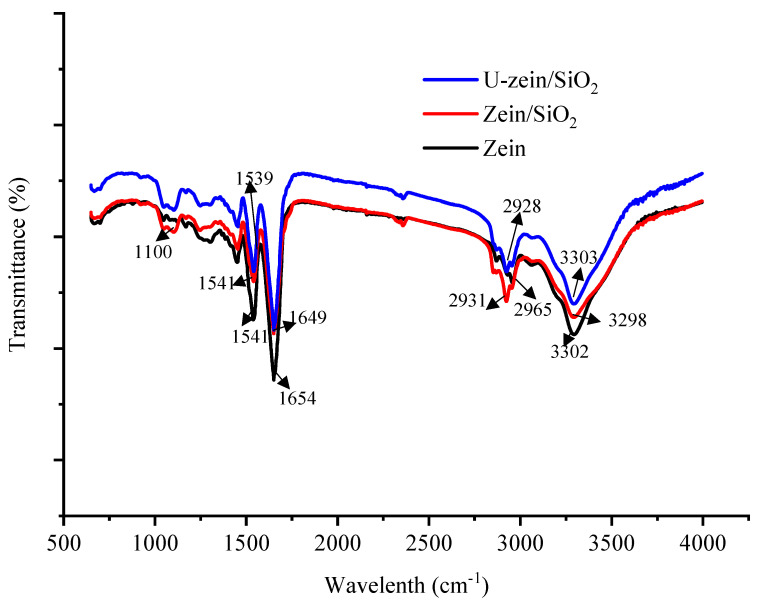
X-ray diffraction spectra of zein-loaded film, zein/SiO_2_ film and U-zein/SiO_2_ film.

**Figure 3 foods-12-03056-f003:**
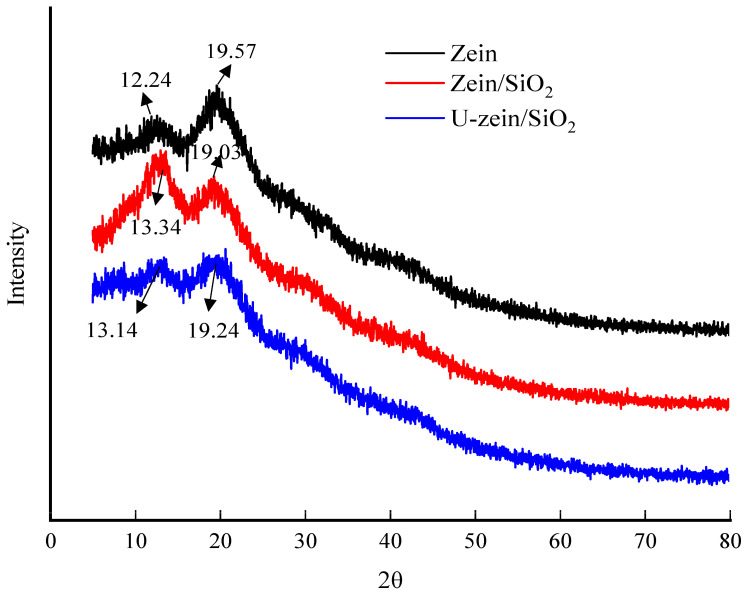
Attenuated total reflectance-Fourier transform infrared spactra of zein-loaded film, zein/SiO_2_ film and U-zein/SiO_2_ film.

**Figure 4 foods-12-03056-f004:**
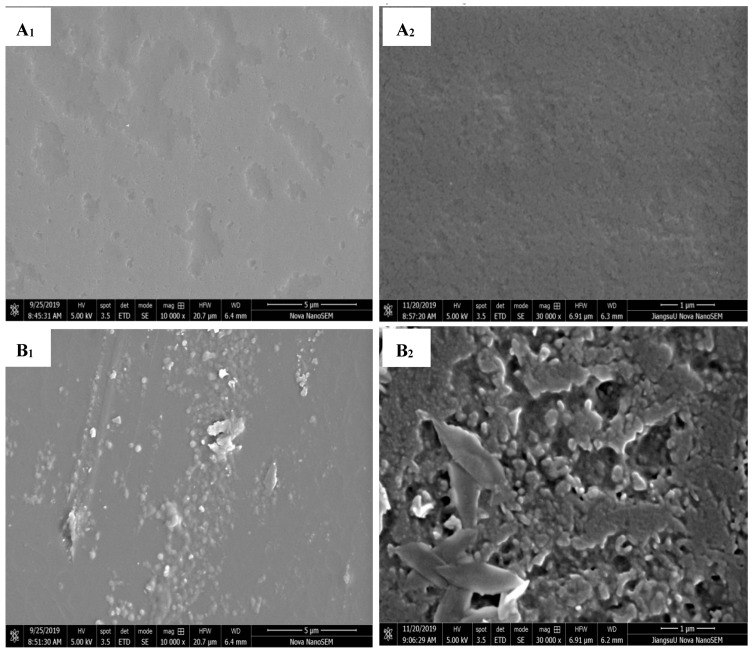
Scanning electron microscopy images of the surface of the film samples with magnifications of 10,000× and 30,000× and the cross section with a magnification of 10,000×, respectively (**A_1_**, **A_2_,** and **A_3_**, zein-loaded film; **B_1_**, **B_2_,** and **B_3_**, zein/SiO_2_ film; **C_1_**, **C_2_,** and **C_3_**, U-zein/SiO_2_ film).

**Figure 5 foods-12-03056-f005:**
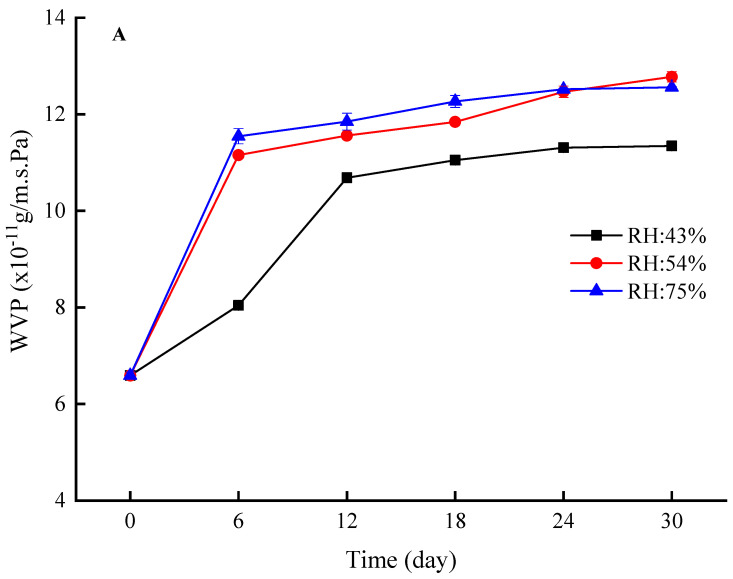
The effects of different relative humidity (RH) on the water vapor permeability (WVP), oxygen permeability, elongation at break (EBA) and tensile strength (TS)of the U-zein/SiO_2_ film during 30 days of storage. (**A**), WVP; (**B**), oxygen permeability; (**C**), EBA; (**D**), TS.

**Figure 6 foods-12-03056-f006:**
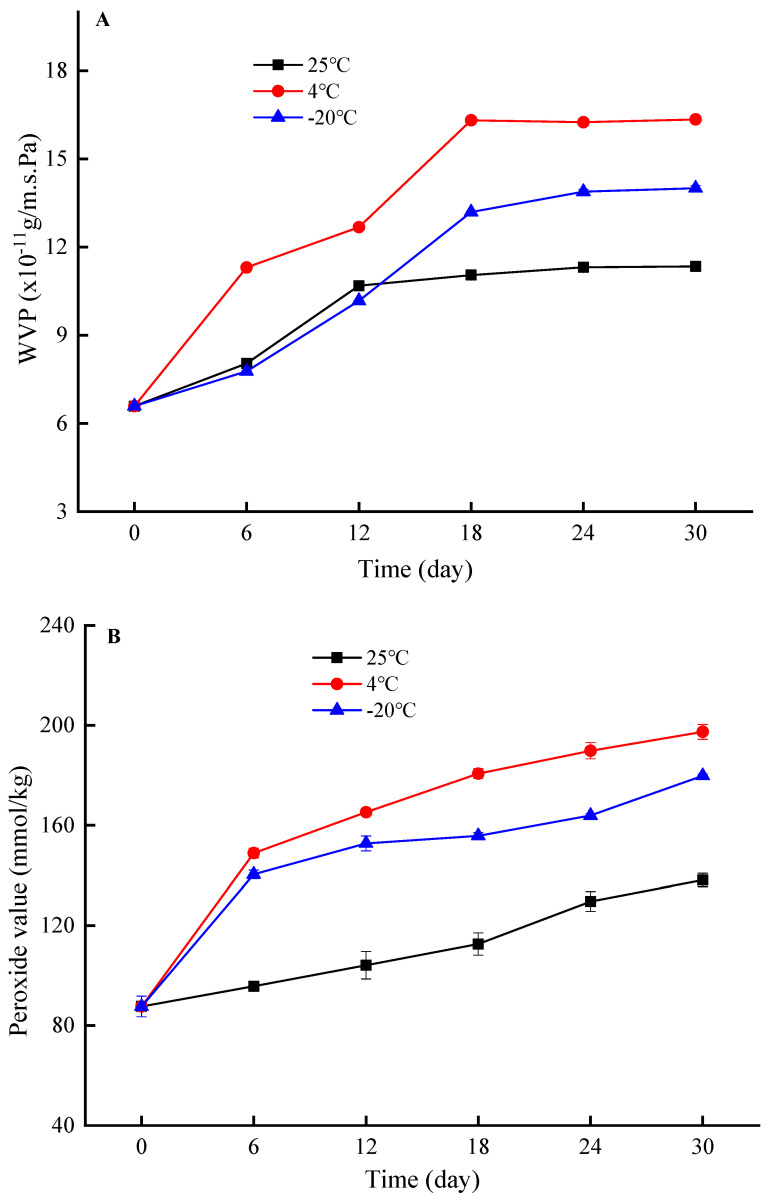
The effects of different ambient temperatures on the water vapor permeability (WVP), oxygen permeability, elongation at break (EBA), and tensile strength (TS) of the U-zein/SiO_2_ film during 30 days of storage. (**A**), WVP; (**B**), oxygen permeability; (**C**), EBA; (**D**), TS.

**Table 1 foods-12-03056-t001:** Physicochemical properties of zein-loaded film, zein/SiO_2_ film, and U-zein/SiO_2_ film.

	Moisture Content (%)	Swelling Degree (%)	Solubility (%)	WVP (×10^−11^ g/m·s·Pa)	Peroxide Value (mmoL/kg)
Zein	17.49 ± 0.13 ^a^	97.67 ± 0.41 ^a^	23.35 ± 0.20 ^b^	13.07 ± 0.34 ^a^	144.97 ± 3.65 ^a^
Zein/SiO_2_	16.42 ± 0.19 ^b^	112.93 ± 0.16 ^a^	24.4 ± 0.09 ^a^	7.21 ± 0.03 ^b^	99.44 ± 1.71 ^b^
U-zein/SiO_2_	15.65 ± 0.18 ^c^	120.65 ± 0.23 ^b^	22.62 ± 0.26 ^c^	6.59 ± 0.07 ^b^	87.64 ± 4.13 ^c^

Different lowercase letters in the same index indicate significant differences between groups (*p* < 0.05).

**Table 2 foods-12-03056-t002:** Relative crystallinity (RC) and thermal properties of zein-loaded film, zein/SiO_2_ film, and U-zein/SiO_2_ film.

	RC (%)	T_0_ (°C)	T_P_ (°C)	T_C_ (°C)	T_C_ − T_0_ (°C)	ΔH (J/g)
Zein	39.09 ± 2.34 ^c^	79.57 ± 1.01 ^a^	204.90 ± 2.31 ^a^	326.38 ± 1.03 ^a^	125.33 ± 0.88 ^a^	178.30 ± 2.14 ^b^
Zein/SiO_2_	52.72 ± 1.92 ^a^	82.36 ± 1.66 ^a^	208.37 ± 1.16 ^ab^	326.35 ± 1.07 ^a^	126.01 ± 5.28 ^a^	183.89 ± 1.15 ^a^
U-zein/SiO_2_	43.51 ± 1.96 ^b^	80.15 ± 2.37 ^a^	219.59 ± 2.42 ^b^	325.48 ± 0.92 ^a^	139.44 ± 3.60 ^a^	180.47 ± 2.35 ^ab^

Different lowercase letters in the same index indicate significant differences between groups (*p* < 0.05).

## Data Availability

The data used to support the findings of this study can be made available by the corresponding author upon request.

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
