# Peer review of "Characterization of Nano-SiO2/Zein Film Prepared Using Ultrasonic Treatment and the Ability of the Prepared Film to Resist Different Storage Environments"

_foods, 2023, doi:10.3390/foods12163056_

Round 1

Reviewer 1 Report

Dear Authors,

The job is very interesting. Corrections are required for:

- In line 315, correct the units (the dots are too large).

- In lines: 337, 395, 410, 416, 428 there are too many punctuation marks - periods.

- Fig. 4 C3 should be placed in the middle, because currently it looks as if there is no picture next to it.

Best Regards

Author Response

The job is very interesting. Corrections are required for:

  1. In line 315, correct the units (the dots are too large).

Answer: We are thankful to the reviewer’s suggestion. Corrected. (Please check line 315)

  1. In lines: 337, 395, 410, 416, 428 there are too many punctuation marks - periods.

Answer: We are thankful to the reviewer’s suggestion. The manuscript has been carefully examined and the extra punctuation has been deleted. Please check the whole manuscript.

  1. Fig. 4 C3 should be placed in the middle, because currently it looks as if there is no picture next to it.

Answer: We are thankful to the reviewer’s suggestion. The Fig. 4 C3 has been placed in the middle. Please check it.

Reviewer 2 Report

The submitted manuscript entitled “Characterization of nano-SiO2/zein film prepared by ultrasonic treatment and the ability of the prepared film to resist different storage environments” is focused on the investigation of zein/SiO2 film prepared by the help of ultrasonic treatment. I find the manuscript up-to-date and successfully designed.

I have some minor comments that should be considered before further processing and acceptance for Foods Journal.

-          line 23:  mistyping error should be corrected (physico-mechanical…)

-          reference (author name) should be edited in line 80

-          check the sentence in lines 119 to 120, if the dot is not missing there

-          the reference to aforementioned procedure should be given (line 135)

-          the sentence is not finished in line 217

-          Figure 1 should be described more precisely, it is not clear from the Figure legend, which film was used for the ultrasound treatment test (lower graph)

-          check the values in line 274 (they are not in accordance with the values in Table 1)

-          Figures 2 and 3 have to be exchanged.

-          SiO2 should be edited with lower index to SiO2

-          successfully should be used in line 373 instead of successful

-          a dot before the reference number should be deleted in line 410 and 416

-          a magnification should be checked in line 444 (3000 vs 30000)

-          considering the Fig. 5C and D, EBA has a decreasing trend, while TS has increased in 30 days time interval. Thus, the last sentence in lines 518 to 520 should be reformulated, in my opinion.

-          the value of POV (138,19 mmol/kg) in line 502 should be checked because it seems that there is a lower final value in the graph (Fig 5B). Moreover, a mentioned value is totally the same as in line 545 (please, check).

Only minor editing in English language is required (listed in comments).

Author Response

The submitted manuscript entitled “Characterization of nano-SiO2/zein film prepared by ultrasonic treatment and the ability of the prepared film to resist different storage environments” is focused on the investigation of zein/SiO2 film prepared by the help of ultrasonic treatment. I find the manuscript up-to-date and successfully designed. I have some minor comments that should be considered before further processing and acceptance for Foods Journal.

  1. line 23: mistyping error should be corrected (physico-mechanical…)

Answer: We are thankful to the reviewer’s suggestion. Corrected. (Please check line 24)

  1. reference (author name) should be edited in line 80

Answer: Corrected. (Please check line 24)

  1. check the sentence in lines 119 to 120, if the dot is not missing there

Answer: We are thankful to the reviewer’s suggestion. The sentence mentioned has been modified to “Zein with a protein purity of 98% was obtained from Sigma-Aldrich (Steinheim, Germany). Nano-SiO2 with purity greater than 96% was purchased from Tianjin Longhua Chemical Co., Ltd.” (Please check lines 119-121)

  1. the reference to aforementioned procedure should be given (line 135)

Answer: We are thankful to the reviewer’s suggestion. The preparation method for the zein-loaded film did not come from the literature, but refers to the preparation method of the zein/SiO2 film mentioned in this section.

  1. the sentence is not finished in line 217

Answer: Corrected. (Please check line 217)

  1. Figure 1 should be described more precisely, it is not clear from the Figure legend, which film was used for the ultrasound treatment test (lower graph)

Answer: We are thankful to the reviewer’s suggestion. In order to make the expression more accurate, the Figure caption has been modified to “The effects of nano-SiO2 concentration (A) on the elongation at break (EBA) and tensile strength (TS) of the composite film without ultrasonic treatment; the effect of ultrasonic treatment time (B) on EBA and TS of composite films.” (Please check lines 257-258)

  1. check the values in line 274 (they are not in accordance with the values in Table 1)

Answer: We are thankful to the reviewer’s suggestion. Corrected. (Please check line 275)

  1. Figures 2 and 3 have to be exchanged.

Answer: We are thankful to the reviewer’s suggestion. Corrected. (Please check the position of the Figure 2 and Figure 3)

  1. SiO2 should be edited with lower index to SiO2

Answer: We are thankful to the reviewer’s suggestion. Corrected. Please check the whole manuscript.

  1. successfully should be used in line 373 instead of successful

Answer: Corrected. (Please check line 273)

  1. a dot before the reference number should be deleted in line 410 and 416

Answer: We are thankful to the reviewer’s suggestion. The manuscript has been carefully examined and the extra punctuation has been deleted. Please check the whole manuscript.

  1. a magnification should be checked in line 444 (3000 vs 30000)

Answer: Corrected. (Please check line 445)

  1. considering the Fig. 5C and D, EBA has a decreasing trend, while TS has increased in 30 days time interval. Thus, the last sentence in lines 518 to 520 should be reformulated, in my opinion.

Answer: We are thankful to the reviewer’s suggestion. It was a mistake in writing. The sentence has been revised as “After 30 days of being stored at an ambient humidity of 54%, the EBA of the film decreased from 108.35% to 88.37%, while its TS increased from 5.53 MPa to 10.87 MPa.” (Please check lines 517-519)

  1. the value of POV (138,19 mmol/kg) in line 502 should be checked because it seems that there is a lower final value in the graph (Fig 5B). Moreover, a mentioned value is totally the same as in line 545 (please, check).

Answer: We are thankful to the reviewer’s suggestion. By carefully checking the data in Fig.5B, the value was corrected to “135.28 mmol/kg” (Please check line 501)